# Impact of Sex in Left Atrial Indices for Prognosis of Heart Failure with Preserved Ejection Fraction

**DOI:** 10.3390/jcm11195910

**Published:** 2022-10-07

**Authors:** Shiro Hoshida, Koichi Tachibana, Nobutaka Masunaga, Yukinori Shinoda, Tomoko Minamisaka, Hirooki Inui, Keisuke Ueno, Masahiro Seo, Masamichi Yano, Takaharu Hayashi, Akito Nakagawa, Yusuke Nakagawa, Shunsuke Tamaki, Takahisa Yamada, Yoshio Yasumura, Yohei Sotomi, Shungo Hikoso, Daisaku Nakatani, Yasushi Sakata

**Affiliations:** 1Department of Cardiovascular Medicine, Yao Municipal Hospital, Yao 581-0069, Japan; 2Division of Cardiology, Osaka General Medical Center, Osaka 558-8558, Japan; 3Division of Cardiology, Osaka Rosai Hospital, Sakai 591-8025, Japan; 4Cardiovascular Division, Osaka Police Hospital, Osaka 543-0035, Japan; 5Division of Cardiology, Amagasaki-Chuo Hospital, Amagasaki 661-0976, Japan; 6Department of Medical Informatics, Osaka University Graduate School of Medicine, Suita 565-0871, Japan; 7Division of Cardiology, Kawanishi City Hospital, Kawanishi 666-0195, Japan; 8Department of Cardiology, Rinku General Medical Center, Izumisano 598-8577, Japan; 9Department of Cardiovascular Medicine, Osaka University Graduate School of Medicine, Suita 565-0871, Japan

**Keywords:** Ed/Ea, Fine–Gray model, HFpEF, LAVI, sex

## Abstract

Objective: We aim to clarify the differences in the association between re-admission for heart failure (HF) and left atrial (LA) overload indices by sex in heart failure and a preserved ejection fraction (HFpEF). Methods: We analyzed 898 HFpEF patients hospitalized for acute decompensated HF. Blood tests and transthoracic echocardiography were performed before discharge. The primary endpoint was re-admission for HF during the first year. Results: The ratio of diastolic elastance to arterial elastance (*p* = 0.014), a relative index of LA pressure overload, in men and LA volume index (LAVI, *p* = 0.020) in women were significant for re-admission for HF during the first year in the multivariable Fine–Gray analysis. Stroke volume (SV)/LA volume (LAV), another index for LAV overload, was not a significant prognostic factor of re-admission for HF during this time. Conclusion: LA overload was an important prognostic factor for HF re-readmission during the first year after enrolment in patients with HFpEF, but the indices relating to LA overload differed by sex.

## 1. Introduction

There is considerable variation in the relationships between the relative indices of left atrial (LA) volume and pressure [1], which could possibly affect medication selection to improve the prognoses of patients with heart failure (HF) and a preserved ejection fraction (HFpEF). Left ventricular (LV) operant diastolic elastance (Ed) and effective arterial elastance (Ea) are easily evaluated using transthoracic echocardiography (TTE) [2,3,4]. We recently reported the differences in the target index of LA overload for each prognostic endpoint in patients with HFpEF [5]. Stroke volume (SV)/LA volume (LAV), a relative index for LAV overload, was a significant prognostic factor of re-admission for HF [5]. Furthermore, we identified sex-based differences in the Ed/Ea ratio in older hypertensive patients without HF [6] and found that the significance of hemodynamic factors for prognosis in patients with HFpEF depends on follow-up duration [7]. Most of the re-admission for HF occurs within the first year after discharge [5]. Sex-based differences in the prognosis are also reported in patients with HFpEF [8]. The prevalence of and hospitalization related to HFpEF are increasing and the growing older population causes further worsening of this trend. Thus, the present study aimed to define the differences in significant prognostic factors by sex during the first year for re-admission for HF using a multivariable Fine–Gray analytical model in patients with HFpEF.

## 2. Methods

We enrolled 898 patients with prognostic data recruited (June 2016–February 2020) from the Prospective Multicenter Observational Study of Patients with Heart Failure with Preserved Ejection Fraction (PURSUIT HFpEF) registry (men/women, 406/492; mean age, 81 years; UMIN ID: UMIN000021831) [9,10] at discharge during the index hospitalization with acute decompensated heart failure. Patients were enrolled based on the Framingham criteria and if they met the criteria of left ventricular ejection fraction ≥50% on TTE and had N-terminal pro-brain natriuretic peptide (NT-proBNP) level of ≥400 pg/mL upon admission. The present study excluded patients with severe aortic stenosis, aortic regurgitation, mitral stenosis, or mitral regurgitation due to structural changes in the valves detected by TTE upon admission. Some patients for whom partial TTE data were available were included.

We previously reported the method for data collection and follow-up/clinical outcome [10]. Survival data were obtained by dedicated coordinators and investigators through direct contact with patients, their physicians at the hospital, or in an outpatient setting, via a telephone interview with their families or by mail. Data collection was performed using an electronic data capture system integrated into the electronic medical records developed at Osaka University [11]. The primary endpoint of this study was re-hospitalization for HF. A HF hospital re-admission was defined as admission to a hospital necessitated by HF and primarily for its treatment. Collaborating hospitals were encouraged to enroll consecutive patients with HFpEF irrespective of treatment. 

Laboratory data, such as levels of creatinine and NT-proBNP, and TTE parameters were examined when patients were stabilized before discharge. Estimated glomerular filtration rate (eGFR) was calculated using the standard method. Blood pressure and heart rate measurements were performed along with the echocardiographic examination and were recorded according to the American Society of Echocardiography or European Society of Echocardiography guidelines [12,13]. Volumetry was standardized using the modified Simpson’s rule. As a relative marker of LA pressure overload for estimating LV diastolic function, we examined the afterload-integrated Ed/Ea ([E/e′]/[0.9 × systolic blood pressure]) [6]. As relative markers of LAV overload, we evaluated the LAV index (LAVI) and the SV/LAV ratio [1]. 

This research was conducted without patient involvement. 

The PURSUIT HFpEF registry was managed in accordance with the principles of the Declaration of Helsinki. The study protocol was approved by the ethics committee of each participating hospital. The protocol (Osaka University Clinical Research Review Committee, R000024414) was approved by the ethics committee of Yao Municipal Hospital (2016-No.0006). All participants provided written informed consent regarding the design and conduct of the study during the indexed hospitalization. We performed only essential examinations in routine clinical practice.

Continuous variables are expressed as mean ± standard deviation, whereas categorical variables are presented as frequencies and percentages. Intergroup differences in categorical variables were assessed using the chi-square test, while those for continuous variables were assessed using Student’s t-test. Each cutoff point of the prognostic factors for re-admission for HF was evaluated using receiver operating characteristic (ROC) curve analysis (Appendix A). To have an insightful interpretation of the impact of sex during the first year, we conducted competing risk analysis. All-cause mortality and re-admission for HF were treated as competing events, and we assumed that patients who died were censored at the date of death. Confounder-adjusted analysis was also conducted by applying the Fine–Gray model [14,15,16], a Cox-type regression analysis for competing risk analysis. A multivariable Fine–Gray analysis using categorical variables determined by each cutoff point was performed by adjusting for comorbidities, including atrial fibrillation, hypertension, diabetes mellitus, dyslipidemia, and history of coronary artery disease. The hazard ratio (HR) with 95% confidence interval (CI) was estimated. 

Statistical significance was set at *p* < 0.05 in general analyses and was set at *p* < 0.1 for an interaction analysis. All statistical analyses were performed using EZR (Saitama Medical Center, Jichi Medical University, Saitama, Japan), a graphical user interface for R (The R Foundation for Statistical Computing, Vienna, Austria).

## 3. Results

A comparison of the clinical and laboratory characteristics and medications between men and women is shown in Table 1. Age, blood pressure, heart rate, incidence of coronary artery disease and hypertension, and use of calcium-channel blockers and renin-angiotensin-aldosterone inhibitors were significantly different between the sexes. However, no differences were observed in the incidence of re-admission for HF during the first year between the sexes (Table 1). Although not shown, 43 (<5%) patients were treated with sodium glucose transporter 2 (SGLT2) inhibitors before discharge. In terms of echocardiographic parameters, the LAVI, LV ejection fraction, E/e′, and Ed/Ea were significantly higher while the SV/LAV ratio, LV dimension, and LV mass index were significantly lower in women than men (Table 2). 

In a comparison of the clinical data between the patients with low and high LAVI before discharge, age, systolic blood pressure, NT-proBNP and eGFR levels, and the incidence of male sex and atrial fibrillation were significantly different between the two groups (Appendix A). In those with high LAVI, larger left ventricle, and higher LV mass index, E/e’ and Ed/Ea were observed than in those with low LAVI (Appendix A).

For re-admission for HF, age, LAVI, Ed/Ea, and eGFR and NT-proBNP levels were all significant factors during the first year in the univariable Fine–Gray analysis (Table 3). SV/LAV (HR 0.632, CI 0.455–0.877, *p* = 0.006) and E/e’ (HR 1.372, CI 1.003–1.876, *p* = 0.048) were also significant prognostic factors during the first year.

In the multivariable Fine–Gray analysis by adjusting for comorbidities in all patients, NT-proBNP level and LAVI, but not age, eGFR level, or Ed/Ea, were significant prognostic factors during the first year in all patients (Table 3). SV/LAV in place of LAVI (HR 0.731, CI 0.487–1.098, *p* = 0.130) or E/e’ in place of Ed/Ea (HR 1.271, CI 0.892–1.810, *p* = 0.180) was not significant for re-admission for HF in the multivariable Fine–Gray analysis. 

When the analysis was performed in each sex, a marked difference was observed between the sexes. In men, Ed/Ea and NT-proBNP level were significant for this endpoint, and LAVI was only a significant prognostic factor in women during the first year (Table 4). When E/e′ was used instead of Ed/Ea in this multivariable model in men, E/e′ was not significant for prognosis (HR 1.368, CI 0.779–2.399, *p* = 0.280). When SV/LAV ratio was used in place of LAVI in the multivariable model, SV/LAV ratio was not significant during the first year in women (HR 0.587, CI 0.324–1.064, *p* = 0.079). In the Fine–Gray model, the interaction for sex was significant only in Ed/Ea (*p* = 0.078, Table 4).

## 4. Discussion

In men, Ed/Ea in association with NT-proBNP level was significant for re-admission for HF during the first year in patients with HFpEF. However, LAVI, but not SV/LAV ratio, was significant for re-admission for HF in women.

In older patients, a reduction in the number of HF admissions may be essential in real-world clinical practice. We previously reported that SV/LAV ratio, but not LAVI, is a significant prognostic factor for HF re-admission during the three years after discharge in patients with HFpEF [5]. LA volume overload resulting in morphological changes in the left atrium relative to SV level is closely related to HF onset. During the first year after enrolment, however, LAVI, but not SV/LAV ratio, was a significant prognostic factor in patients with HFpEF, especially in women, although between-sex interaction was not significant in the case of LAVI for HF re-admission. In patients with high LAVI, sufficient diuretic use may be essential to avoiding a volume shift to the third space of the body, resulting in the prevention of an HF re-admission during this phase.

Once the left atrium is enlarged, it does not shrink even after a substantial volume reduction. Any disease except for certain valvular diseases does not induce only LA volume or pressure overload. LAV is an indicator of a long-term LV filling pressure elevation, and a morphologically enlarged LAV may be a secondary phenomenon. The cardiothoracic ratio and LAVI in patients with HFpEF are larger in women than in men [7], although their LV dimensions are significantly smaller. Since LAVI, but not SV/LAV ratio, was a significant prognostic factor during the first year only in women, LA enlargement is likely to occur to a larger extent in women than in men. LAVI itself may be a more potent determining factor for HF re-admission in women during the short-term period than SV/LAV ratio, possibly leading to the differences in LAV-related prognostic factors for HF re-admission between short- and middle-term durations. 

In contrast, it may be difficult to enlarge the left atrium during the short-term period in men, leading to higher LA pressure, resulting in the finding that Ed/Ea, a relative index for LA pressure overload, was a determining factor for HF re-admission during that time in men. A positive interaction of Ed/Ea between sexes for HF re-admission strongly suggests this issue. In other words, apparent LA enlargement may not be a useful marker for HF re-admission in men.

We included laboratory data in the multivariable Fine–Gray model by adjusting for comorbidities and revealed a significant association with NT-proBNP level in men, but not in women, during the first year in the case of re-admission for HF. The reason for this between-sex difference is as yet unclear. NT-proBNP level may reflect the extent of LA pressure overload more than that of LA volume overload if LV performance is the same. Since LV enlargement might be a long-term change in HFpEF, the time from first diagnosis of HF or structural heart disease to index hospitalization may be taken into account in the analysis.

Current treatment strategies in HFpEF are limited, but there are now promising results using SGLT2 inhibitors that show significantly reduced cardiac death or hospitalization for HF [17,18]. However, one should pay attention to the differences in age of participants included in the study (10 years older in our study than in [17]) and the between-sex differences in the effect on prognosis. We are waiting for good results of ongoing large-scale studies in patients with HFpEF, where desirable treatment provided could effectively reduce Ed/Ea in men and LAVI in women in association with the reduction in HF re-admission.

## 5. Conclusions

LA overload was an important factor for HF re-readmission in older patients with HFpEF, although the indices for LA overload differed by sex during the first year.

## Figures and Tables

**Table 1 jcm-11-05910-t001:** Between-sex differences in patient characteristics before discharge and event rates.

		Men (*n* = 406)	Women (*n* = 492)	*p* Value
Age, years		80 ± 9	82 ± 8	<0.001
Systolic blood pressure, mmHg		122 ± 19	120 ± 19	0.044
Diastolic blood pressure, mmHg		64 ± 12	66 ± 12	0.017
Heart rate, bpm		68 ± 15	72 ± 15	<0.001
Log (NT-proBNP)		3.07 ± 0.53	3.04 ± 0.51	0.465
eGFR, mL/min/1.73m^2^		43.4 ± 19.6	42.6 ± 19.0	0.516
Atrial fibrillation, *n* (%)		166 (41)	179 (36)	0.167
Coronary artery disease, *n* (%)		108 (27)	66 (14)	<0.001
Diabetes mellitus, *n* (%)		145 (36)	151 (31)	0.111
Dyslipidemia, *n* (%)		160 (40)	209 (43)	0.351
Hypertension, *n* (%)		355 (88)	406 (83)	0.041
Medications				
Beta-blockers, *n* (%)		222 (55)	277 (56)	0.626
Calcium-channel blockers, *n* (%)		224 (55)	215 (44)	<0.001
Diuretics, *n* (%)		338 (83)	398 (81)	0.361
RAAS inhibitors, *n* (%)		313 (77)	341 (69)	0.009
Statins, *n* (%)		131 (32)	165 (34)	0.686
Re-admission for HF				
During years 0-1, *n* (%)		83 (20)	106 (22)	0.687

Values are mean ± standard deviation or number (%). eGFR, estimated glomerular filtration rate; HF, heart failure; NT-proBNP, N-terminal pro-brain natriuretic peptide; RAAS, renin-angiotensin-aldosterone system.

**Table 2 jcm-11-05910-t002:** Between-sex differences in echocardiographic data before discharge.

		Men	Women	*p* Value
LAD, mm		45 ± 8	44 ± 9	0.125
LAVI, mL/m^2^		52 ± 25	58 ± 32	0.014
SV/LAV		0.778 ± 0.379	0.652 ± 0.355	<0.001
LVDs, mm		32 ± 6	28 ± 5	<0.001
LVDd, mm		48 ± 6	44 ± 6	<0.001
LVEF, %		59.7 ± 7.7	61.2 ± 7.9	0.004
LVMI, g/m^2^		111.8 ± 33.6	104.3 ± 35.4	0.001
DcT of E wave, sec		0.21 ± 0.06	0.22 ± 0.07	0.262
E/e’		12.6 ± 5.1	15.2 ± 6.9	<0.001
Ed/Ea		0.115 ± 0.049	0.143 ± 0.069	<0.001

Values are mean ± standard deviation. DcT, deceleration time; E, early transmitral flow velocity; e’, onset of early diastolic mitral annular velocity; Ea, arterial elastance; Ed diastolic elastance; LAD, left atrial diameter; LAV, left atrial volume; LAVI, left atrial volume index; LVDs, left ventricular end-systolic dimension; LVDd, left ventricular end-diastolic dimension; LVEF, left ventricular ejection fraction; LVMI, left ventricular mass index; SV, stroke volume.

**Table 3 jcm-11-05910-t003:** Univariable and multivariable Fine–Gray analyses in re-admission for heart failure during the first year in all patients.

	Univariable			Multivariable		
	Ratio	95% CI	*p* Value	Ratio	95% CI	*p* Value
Age	1.421	1.017–1.986	0.039	1.144	0.753–1.737	0.530
LAVI	2.069	1.438–2.978	<0.001	1.541	1.004–2.365	0.048
Ed/Ea	1.447	1.012–2.069	0.043	1.406	0.941–2.100	0.096
eGFR	0.476	0.342–0.664	<0.001	0.747	0.488–1.142	0.180
NT-proBNP	2.564	1.742–3.774	<0.001	1.867	1.184–2.946	0.007

CI, confidence interval; Ea, arterial elastance; Ed, diastolic elastance; eGFR, estimated glomerular filtration rate; LAVI, left atrial volume index; NT-proBNP, N-terminal pro-brain natriuretic peptide.

**Table 4 jcm-11-05910-t004:** Between-sex differences in prognostic variables for re-admission for heart failure during the first year in a multivariable Fine–Gray analysis.

							Interaction
	Men			Women			Men vs. Women
	Ratio	95% CI	*p* Value	Ratio	95% CI	*p* Value	*p* Value
Age	1.272	0.692–2.338	0.440	1.061	0.602–1.867	0.840	0.860
LAVI	1.251	0.699–2.236	0.450	2.094	1.121–3.912	0.020	0.190
Ed/Ea	2.108	1.161–3.825	0.014	0.965	0.558–1.669	0.900	0.078
eGFR	0.810	0.446–1.471	0.490	0.694	0.376–1.281	0.240	0.670
NT-proBNP	2.462	1.214–4.994	0.012	1.643	0.900–2.999	0.110	0.290

CI, confidence interval; Ea, arterial elastance; Ed, diastolic elastance; eGFR, estimated glomerular filtration rate; LAVI, left atrial volume index; NT-proBNP, N-terminal pro-brain natriuretic peptide.

## Data Availability

Data are available on reasonable request.

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
