# Peer review of "Impact of Sex in Left Atrial Indices for Prognosis of Heart Failure with Preserved Ejection Fraction"

_jcm, 2022, doi:10.3390/jcm11195910_

Round 1
Reviewer 1 Report
This report / document by Hoshida et al. points out the importance of a decrease in the number of individuals hospitalized for HF and it is well known that LA overload is a crucial factor for re-admission in older patients with HFpEF.
In this study it has been demonstrated that the indices for LA overload differed by sex during the first year, in particular Ed/Ea (diastolic elastance/arterial elastance), in association with NT-proBNP level, was significant for re-admission of HF in men, but not in women. On the other hand, LA volume index (LAVI) was an important prognostic variable only in women.
The survey is well structured, starting from the patient’s enrolment and its exclusion criteria. The methods are well described and the results are interpreted in an appropriate way, supporting the conclusions. Moreover, it is remarkable the presentation of the results, which is clear and concise due to an accurate use of the Tables.
The manuscript is original and it might improve the clinical practice thanks to the differences highlighted between males and females. The treatment might be designed to reduce Ed/Ea in men and LAVI in women, leading to the decrease of hospitalizations for HF. Considering all above, further studies will be necessary to decide the most suitable therapy.
Author Response
Thank you for your nice comments. As the reviewer pointed out, we are waiting for good results of ongoing large-scale studies in patients with HFpEF treated with pharmacological agents such as SGLT2 inhibitors (page 6, lines 194-198).

Reviewer 2 Report
A very nice study based on carefully collected data from patients with atrial fibrillation (AF). The authors investigated which of the various echocardiographic and laboratory parameters could serve as predictors. They found that readmission for AF depended on several left atrial congestion parameters. Ed/Ea in men and LAVI in women. While it is clear that left atrial overload is a predictor of AF recurrence and therefore readmission, it is interesting to know that the parameters being studied is different in men and women. It would be good to know, perhaps using ROC curves to analyse sensitivity and specificity, how well these parameters really serve as predictors. Perhaps thresholds can even be established so that the outpatient cardiologist can assess how aggressively he/she should treat LA-overlad with drugs such as flecainide or others. I was quite surprised at the age of the patients. The vast majority were octogenarians. Is this a problem specific to the ageing Japanese population? Here in Europe, these patients seem to be younger when they become visible with atrial fibrillation.
Author Response
Thank you for your suggestions. We investigated which of the various echocardiographic and laboratory parameters could serve as predictors for re-admission for heart failure (HF). We found that readmission for HF depended on several LA congestion parameters: Ed/Ea in men and LAVI in women. We clearly showed that left atrial overload is a predictor of HF re-admission. We performed ROC curve analysis for detecting a cut-off value of each variable of echocardiographic and laboratory examination in all patients together, but not in each sex separately. As the reviewer pointed out, it is important thresholds could be established so that the outpatient cardiologist can assess how aggressively he/she should treat LA-overload with drugs. In this sence, Ed/Ea >0.097 (Hoshida S, et al. ESC Heart Failure 2022) and LAVI >43 mL/m2 (Suppl. Table 1, this study) may be important in clinical setting, although the prospective study is needed to clarify this issue. In real-world Japanese population, the mean age of HFpEF patients admitted to hospitals is more than 80 years.

Round 2
Reviewer 2 Report
Thank your for the clarifications and adjustments. Please provide the ROC-Curves you mentioned in the supplementary material
Author Response
We provided the ROC curves of LAVI and Ed/Ea in the Suppl. Figure 1 (page 2, line 96).
